# ADVERSARIAL AGENT COLLABORATION FOR C TO RUST TRANSLATION

## ABSTRACT

Translating C to memory-safe languages, like Rust, prevents critical memory safety vulnerabilities that are prevalent in legacy C software. Existing approaches for C to safe Rust translation, including LLM-assisted ones, do not generalize on larger ($> 500$ LoC) C codebases because they depend on complex program analyses that frequently break. In this work, we present ACToR (**A**dversarial **C To R**ust translator), a simple LLM agent-based approach. Inspired by GANs, ACToR pits a generator agent against a discriminator agent, which collaborate to iteratively generate a Rust translation. On each iteration, the translator agent synthesizes and refines a Rust translation to pass an existing suite of tests, and then the discriminator agent finds new failing tests. We demonstrate that ACToR translates all of the 63 real-world command-line utilities considered in our benchmarks, which have an average size of 473 lines of code, and it achieves over 90% test pass rate with zero human intervention during translation. To our knowledge, it is the first work to show evidence that an agent-centric approach can reliably and automatically convert standalone command-line C programs at this scale. Furthermore, ACToR improves translation correctness by up to 25.1% compared to baseline, non-adversarial approaches.

## 1 INTRODUCTION

Memory safety vulnerabilities in C/C++ code constitue a large fraction of security vulnerabilities each year. Microsoft reported 70% of their CVEs are due to memory safety issues (Microsoft, 2019a;b). In response, there has been an increasing demand to translate memory-unsafe legacy code to modern memory-safe languages like Rust from industry and governments (DARPA, 2025), thereby ensuring full memory safety by design. Unfortunately, given the scale of legacy code that permeates our software systems today, manual translation of (millions of lines of) existing C code is a formidable and practically infeasible task. A reasonable translation from C to Rust must use safe and idiomatic Rust programming abstractions, which are checked mostly at compile time and avoid runtime performance overhead, unlike memory safety checking directly in C (Nagarakatte et al., 2010; 2009; Orthen et al., 2024; Serebryany et al., 2012).

Existing approaches for automated translation come in two flavors: rule-based and LLM-assisted. Rule-based translators, like c2rust (Immunant) and its derivatives (Zhang et al., 2023; Emre et al., 2023; Ling et al., 2022), produce unidiomatic and unsafe rust, which limits their usefulness in practice. LLM-assisted translators produce more idiomatic and safe rust, however current works along this line fail to generalize to large ($> 500$ lines of code) C projects. No prior work on LLM-assisted translation has demonstrated their approach successfully translating more than one large C project, and several report requiring manual intervention owing to this brittleness (Shetty et al., 2024; Cai et al., 2025). This brittleness is largely due to their dependency on complex program analyses that break on unseen C programs.

In this work, we aim to advance the state-of-the-art in automated translation and turn our attention towards LLM agents. LLM agents have shown promising capabilities in solving generic but complex software engineering tasks (Jimenez et al., 2023) owing to their ability to iteratively interact with development environments, utilizing tools like code editors, compilers, and unit testing to detect and correct errors to improve results (Chen et al., 2023; Dong et al., 2024; Huang et al., 2023; Qian et al., 2024). In the context of automated translation, a straightforward agentic setup is as follows —-

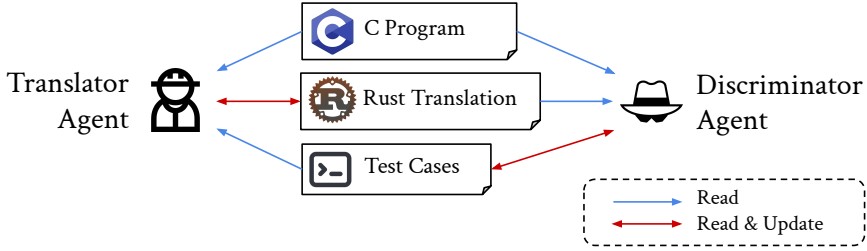

Figure 1: High-level overview of ACToR. The Translator and the Discriminator agents update the translation and the tests in turn to iteratively improve the correctness of the translated program.

the agent has access to the development environment including the Rust compiler, and proceeds by translating the C code to Rust, and iteratively refines it by incorporating feedback from the compiler (in case the translated Rust program does not compile) or from the test inputs on which the translated code provides an output that differs from the output of the original C program. Test suites serve as a useful and pragmatic proxy for the more rigorous, but largely elusive and impractical alternative of full-fledged formal verification for semantic equivalence between the cross-language code, and have unanimously been deployed in prior works (Yang et al., 2024; Eniser et al., 2024; Wang et al., 2025; Zhang et al., 2025).

While the ability to access development environment, without an a priori fixed workflow is a promising first step towards more reliable and realistic translations, we find, rather unsurprisingly, that simplistic code agents like the one described above nevertheless produce incorrect translations; the behavior of the translated Rust code they produce diverges significantly from that of the original C code on inputs absent from the test suite. As a result, such a vanilla agentic setup requires frequent developer intervention and offers little to no improvement over traditional LLM-based translation techniques. At a high level, a simplistic agentic setup, like the one we just described, lacks the key ability of *generalization* — it produces a translation that is overfitted to the (typically small) test suite available to it, but often cannot ensure expected behavior on inputs absent from the suite. Unfortunately, as our evaluation shows, simple fixes such as by increasing the test coverage on the C source, which we achieve by augmenting agents with standard test coverage tools (GNU, 2025), continue to remain ineffective in enhancing the quality of translations.

In this work, we introduce a new agentic setup for program translation designed to overcome the limited generalization observed in simpler agent frameworks. Inspired by the generator–discriminator paradigm popularized by GANs (Goodfellow et al., 2020), we employ two collaborating agents: a *translator*, which proposes candidate translations from C to Rust, and a *discriminator*, which actively searches for evidence( in the form of counterexamples) that the translation diverges from the original. Whenever the discriminator identifies such adversarial inputs, these are fed back to the translator, which uses them to refine subsequent translations. Through this iterative process, the translator learns to produce Rust programs that not only pass the original test cases but also withstand adversarial scrutiny, leading to translations that are more robust and semantically faithful to the original C program. We refer to this adversarial translator–discriminator framework as **ACToR**, for Adversarial C To Rust, as demonstrated in Figure 1.

We demonstrate the effectiveness of ACToR on two sets of benchmarks: A micro-benchmark set of 6 C programs considered in recent work, and, a much larger macro-benchmark set of 57 C programs from the standard BSDCoreUtils. These 63 real-world C programs range from 7 to 5,469 lines of code (LoC), with a median of 473, and totaling around 30,000. **ACToR automatically produces Rust translations for all benchmarks that pass more than 90% of tests on average with zero human intervention during translation.** To our knowledge, it is the first work to show evidence that an agent-centric approach can reliably and automatically convert standalone command-line C programs at this scale. In addition, we show that ACToR's adversarial design choice improves pass rates by up to 25.1% compared with the baseline, and it also improves the quality of tests generated, as measured by their ability to find discriminating behavior.

## 2 RELATED WORK

Automated translation of C to Rust is a relatively recent approach towards achieving full memory safety (Emre et al., 2021). The desired goal of such translation is to generate safe[1] Rust code corresponding C code, i.e., Rust code that uses only idiomatic Rust primitives that the compiler can statically verify or trust to be memory safe. While several prior works have tried, none have worked reliably even for programs of moderate size (e.g. about 500 LoC) for varied reasons (Li et al., 2024).

### 2.1 PROGRAM TRANSLATION (C TO RUST)

Existing works on C-to-Rust translation can primarily be categorized as rule-based or LLM-based techniques. Rule-based techniques such as c2rust (Immunant) and corrode (Sharp) opt for mechanical, line by line, rewriting of the C code, often resorting to raw pointers and unsafe Rust. Later efforts explored ways to infer Rust types from C code, either by compiler feedback (Emre et al., 2023; Hong & Ryu, 2025b), SMT solving (Zhang et al., 2023), data-flow graphs (Xu & Huang, 2025), or syntactic rewriting (Ling et al., 2022), and specialized analyses targeted idioms like tagged unions (Hong & Ryu, 2024b), output parameters (Hong & Ryu, 2024a), and file APIs (Hong & Ryu, 2025a). Beyond general C, Low* compilation to Rust (Fromherz & Protzenko, 2024) demonstrates scalable translation in restricted settings, though such techniques remain infeasible for general-purpose C due to expressiveness and unmodeled APIs.

LLM-based approaches use an LLM to propose a translation (Lachaux et al., 2020). Recent work augments models with different kinds of tools. These include test generation tools (Eniser et al., 2024; Yang et al., 2024), dynamic pointer analyses (Shetty et al., 2024), or multi-modal prompting (Nitin et al., 2024). Function-level decomposition has been explored to improve scalability (Shiraishi & Shinagawa, 2024; Cai et al., 2025; Ou et al., 2025), while agentic workflows combine LLMs with static analysis (Zhou et al., 2025; Farrukh et al., 2025). Neural methods yield more idiomatic code, yet often fall short of preserving exact input/output behavior. Moreover, current work fails to generalize on larger C programs. Studies echo these concerns: decomposition introduces inconsistencies (Li et al., 2024), and LLMs remain prone to subtle errors (pan, 2024).

Work beyond C-to-Rust highlights related strategies. Go-to-Rust translation explored scalable decomposition (Zhang et al., 2025); TransMap (Wang et al., 2023b) and DuoGlot (Wang et al., 2023a) used semantic checking or guided search; skeleton-based methods modeled programs as composable fragments (Wang et al., 2025). Classical efforts translated across other pairs (C++→Java (Buddrus & Schödel, 1998; Malabarba et al., 1999; Beevi et al., 2014), Java→Python (Lachaux et al., 2020; Ibrahimzada et al., 2025)). Meanwhile, verified lifting has advanced in domain-specific contexts, including numerical kernels (Kamil et al., 2016), tensor operators (Zhan et al., 2024; Qiu et al., 2024), and grammar-driven rewriting (Cordy, 2006), with early LLM-based attempts (Bhatia et al., 2024). These exemplify the correctness ideal but are not yet generalizable to C.

Overall, there is growing consensus in favor of techniques that combine rule-based precision with neural readability. Though, to the best of our knowledge, none of the existing techniques is automatic—they either primarily work on small pieces of code or have a complex implementation, often requiring human interventions when aiming for functioning translations of whole C programs. In contrast, a prime focus of our work is the design of a simple framework that leverages LLM agents for C to Rust translation which is easier to implement but also robust enough to automatically translate whole programs to functioning safe Rust code.

### 2.2 COLLABORATIVE LLMS & AGENTS

Many works propose mutli-LLM and multi-agent systems, where the LLMs or agents collaborate to produce a final answer. Many works on code generation propose specific roles, such as coder, tester, or analyst, which collaborate to write code (McAleese et al., 2024; Chen et al., 2023; Dong et al., 2024; Huang et al., 2023; Qian et al., 2024; Islam et al., 2024). Collaborative approaches have been used outside of code generation as well. Many works propose self-reflection techniques (Shinn et al., 2023; Madaan et al., 2023; Dhuliawala et al., 2023), sometimes adied by tools (pan, 2024;

---

[1]The use of `unsafe` keyword in Rust allows mixing unchecked and possibly memory unsafe code in otherwise safe Rust

Gou et al., 2024; Dong et al., 2025), where the LLM first proposes an answer, then reflects on its answer to improve accuracy. Similar works propose multi-agent debate architectures (Chan et al., 2024; Du et al., 2023; Liang et al., 2023; Hong et al., 2024; Wu et al., 2024; Liu et al., 2024) where mutliple answers are proposed, then the LLM(s) debate the correct answer. Several works propose frameworks for agent collaboration (Hong et al., 2024; Wu et al., 2024; Liu et al., 2024).

The key difference between ours and prior work is in the high-level principle: To reliably improve the translated code, we leverage the C program as the *oracle* in the construction of a discriminator, which automatically identifies correctness gaps and provide iterative feedback to the translator.

## 3 ACToR DESIGN

We consider a realistic setup where C project is required to be translated to safe Rust. We explain how our adversarial translation approach solves this task with no human intervention beyond providing the original C program and a set of initial seed test cases.

### 3.1 CODE TRANSLATION TASK

Given a source C program $c$, the goal of the translation is to find a *functionally correct* and *safe* Rust code $rs$ that has equivalent external behaviors as the source C program on all valid inputs. External behaviors include standard output, error messages, file modifications, and so on. More specifically, we want the translated code to satisfy the boolean requirement $\texttt{IsEq}(c, rs)$, where $\texttt{IsEq}$ is an oracle that takes a C program and a Rust program as input, and determines whether they have equivalent behaviors on the valid universe $U$ of inputs that do not violate memory safety. This objective naturally captures the desired requirement that the translated Rust program has the same behavior as the C program on safe inputs. On the other hand, memory-safe handling of malicious inputs is guaranteed by the safe Rust compiler itself.

A naive approach to obtaining a Rust translation is to prompt the LLM agent with the C code $c$ and a small finite set $T \subset U$ of inputs sampled from the safe input universe $U$ with the goal of computing a Rust program that matches the behaviour of $c$ on all inputs from the sampled set $T$. Unfortunately, such a naive method does not produce a satisfactory translation. The space of valid inputs is typically infinite for most programs. Indeed, we emprically observed that with only a limited number of input instances tested one-shot by the agents, the produced translation does not generalize to unseen test cases beyond the limited set $T$.

We remark that a perfect oracle $\texttt{IsEq}$ does not exist in practice. There is no automated tool that practically works in determining whether the given C code and Rust are equivalent in the input universe, or provide counterexamples when they mismatch. On the contrary, what we have is '$\texttt{IsEq*}$', which takes as input both the C and the Rust code, together with a *concrete input* $t$, tells whether they mismatch on this input $t$. This intuitively means we run the C code and Rust code on the same input, and check if they result in the same effect. The problem now becomes how to satisfy $\texttt{IsEq}$ with only $\texttt{IsEq*}$ available.

The next most natural solution is to continually sample new test cases from the input universe $U$ and use them to improve the translated code iteratively. However, this approach is not particularly efficient and does not help much in terms of improving the equivalence on $U$. As the number of tests taken increases, the probability of sampling an input that re-explores a previously tested behavior increases. The incremental improvement per new test becomes modest as we proceed.

### 3.2 ADVERSARIAL TRANSLATION

We propose ACToR, which leverages the adversarial collaboration between two agents. The idea behind ACToR comes from GAN, where two opponents are included to estimate a data distribution implicitly. ACToR consists of two collaborative agents: the translator agent and the discriminator agent. The discriminator aims to surface the mismatches in translated code, while the translator aims to improve the translation to evade detection by the discriminator. The two agents work in turn, iteratively improving the translation. The detailed algorithm is shown in the Algorithm 1.

---

**Algorithm 1** ACToR: Adversarial C to Rust Translation

---

**Input:** C program $c$, seed test set $T_0$, test input universe $U$ (implicit)
**Params:** MaxIter (of outer loop), TestBatchSize, RetryLimit (of inner loops)
**Output:** Final translation $rs^*$ and final test set $T^*$
1: **while** NOTPASSESSUITE$(c, rs, T)$ **do**                    ▷ initial translation on seed tests
2:     $rs \leftarrow$ TRANSLATOR$(c, \varnothing, T_0)$
3: **end while**
4: **assert** ISSAFE$(rs)$
5: $T \leftarrow T_0$
6: **for** $k = 1$ **to** MaxIter **do**
7:     $numRetry \leftarrow 0$
8:     **repeat**
9:         $Batch \leftarrow$ DISCRIMINATOR$(c, rs, n = \text{TestBatchSize})$
10:        $IsValid \leftarrow \big(\forall t \in Batch : \text{VALIDTESTFORMAT}(t) \wedge \text{SANITYOK}(t, c)\big)$
11:        $IsDisc \leftarrow \big(\exists t \in Batch : \neg \text{ISEQ}^*(c, rs, t)\big)$
12:        $numRetry \leftarrow numRetry + 1$
13:        $IsReachingLimit \leftarrow (numRetry \geq \text{RetryLimit})$
14:    **until** $(|Batch| = \text{TestBatchSize} \wedge IsValid \wedge IsDisc) \vee IsReachingLimit$
15:    $T \leftarrow T \cup Batch$
16:    $numRetry \leftarrow 0$
17:    **while** NOT PASSESSUITE$(c, rs, T) \wedge numRetry \leq \text{RetryLimit}$ **do**
18:        $numRetry \leftarrow numRetry + 1$
19:        $rs \leftarrow$ TRANSLATOR$(c, rs, T)$
20:    **end while**
21:    **assert** ISSAFE$(rs)$
22: **end for**
23: **return** $rs$ as $rs^*$, $T$ as $T^*$

---

ACToR takes as input the C code $c$, the universe $U$, the seed test set $T_0$, together with hyperparameters such as the number of iterations the translator and discriminator interact (MaxIter) before ACToR outputs the final translated program. ACToR then conducts iterative translation on the source C code and outputs both the Rust translation $rs*$ and the final test set $T*$. During iterative improvement, ACToR maintains an append-only test set $T$. In the beginning, the translator agent generates the initial safe translation and ensures it passes *all the seed tests* (lines 1-3) before the adversarial iterative loop (involving the discriminator) begins. Next, in each iteration that follows, the discriminator agent reads the C code $c$ and the current Rust code $rs$, and generates test cases on which C and Rust outputs disagree (lines 7-13). The discriminator is structured so that (1) all the test cases abide by the correct format and also pass basic sanity checks when running them on the source C program (line 9); and that (2) some test cases witness a mismatch between outputs of the C and the Rust code that the translator has created so far (line 10). We ensure this by providing a prompt to the discriminator to continue retrying until the generated test cases meet the above two requirements. Once successful, the newly generated test cases ($Batch$) are added to the full test set. The translator agent then takes the updated test cases as guidance to improve the translation. It is set up with prompts that repeatedly ask it to pass all the test cases in the current test set and only use fully safe Rust (lines 16-21). If the translator runs out of retries but still cannot fully pass the test set $T$, we let it skip and move to the next iteration. After a fixed maximum number of iterations, ACToR stops the loop and outputs the final translation and the test set generated.

As the iteration progresses, it becomes increasingly challenging for the discriminator agent to identify differences solely by examining the source code or conducting careful manual testing. To increase the limit, we design a simple agent-friendly fuzzing script to accelerate the mismatch discovery. With the help of a fuzzer script, the discriminator gains more power in finding errors in corner cases. We show the effectiveness of the fuzzing-augmented discriminator embodied in ACToR in evaluations.

## 4 EVALUATION

The goal of a good defense is to retain the functionality of the given C code, and make the defended program fully memory-safe. We evaluate our ACToR-generated Rust code on these rubrics.

Specifically, we evaluate ACToR on the following criteria:

1. **Scalability.** What size of C programs can ACToR automatically translate?

2. **Correctness.** Do the Rust translations maintain equivalent behavior to the C program? Do the Rust translations use safe Rust?

3. **Ablation.** How much does the adversarial agentic design contribute to correctness? How do different experiment configurations influence the final results?

## 4.1 EXPERIMENT SETTINGS

Table 1: Benchmark Statistics

(a) Micro Benchmark

| Program | LOC | Funcs | Cov. |
|---------|-----|-------|------|
| printf | 371 | 11 | 95% |
| expr | 452 | 17 | 92% |
| fmt | 416 | 12 | 91% |
| test | 528 | 17 | 78% |
| join | 475 | 10 | 90% |
| csplit | 303 | 7 | 88% |
| **Total** | **2545** | **74** | **89%** |

(b) Macro Benchmark by Category

| Category | LOC | Funcs | Count |
|----------|-----|-------|-------|
| Text Processing | 13269 | 355 | 21 |
| File Dir. Mgmt. | 6683 | 151 | 15 |
| Scripting Utils | 4204 | 118 | 8 |
| Sys. Info Status | 2908 | 70 | 10 |
| Env. Proc. Ctrl. | 204 | 8 | 3 |
| **Total** | **27268** | **702** | **57** |

**Benchmarks.** We use two sets of benchmarks in our evaluation: a micro-benchmark set and a macro-benchmark set. The micro-benchmark comes from a prior user study on C to Rust translation, which contains 6 standalone C programs, with an average of 424 lines of code each. We manually craft at least 60 test cases for each program in the micro benchmark with an average of 89% line coverage, which are used to evaluate the correctness of translations. We use this small-scale benchmark to demonstrate ACToR's ability to adapt to different LLMs and coding agent frameworks, and to perform an ablation study.

Our main macro-benchmark is a set of 57[2] real-world system utility programs (e.g., `ls`, `cp`, `xargs`) obtained from a software project named BSDCoreUtils. It has an average of 478 lines of code each. We use this benchmark to evaluate scalability of ACToR, and compare correctness to our baselines. Given the large number of programs, manually crafting a comprehensive set of test cases for each macro-benchmark is infeasible. Moreover, automated test generation techniques can produce test sets that unfairly advantage a specific translation approach. Instead, we perform a relative evaluation between ACToR and baselines, as described below in the Evaluation Metrics section.

To minimize measurement noise and make our experiments automatable, our evaluation focuses on single-threaded C programs that have deterministic behaviors across different executions, which is currently an assumption of our testing framework that both the translator and the discriminator depends on. Table 1 summarizes the statistics of two benchmarks, where we categorize the 57 programs into five categories: process execution control, file/directory management, shell scripting commands, system information access, and textural content processing.

**Agent Frameworks and Models.** We use Claude Code (Anthropic, 2025a) with Claude Sonnet 4.5 (Anthropic, 2025b) as the main agent framework and LLM for our experiments. To test if our approach generalizes to other agent frameworks and LLMs, we also performed experiments using Mini-SWE-Agent (SWE-agent, 2025), a popular open-source agent framework, with GPT 5 mini (OpenAI, 2025), Claude 4 Sonnet, and Claude 4.5 Sonnet as the LLMs. We use 'ClaudeCode+Sonnet4.5', 'ClaudeCode+Sonnet4', 'SWE+Sonnet4.5', 'SWE+Sonnet4', and 'SWE+GPT5mini' to represent them in later sections, respectively.

**Comparison Baselines.** We compare ACToR with two non-adversarial baselines, and one adversarial LLM agent baseline. (1) *Naive baseline:* a single agent instructed to translate the C program using

---

[2]The full set of BSDCoreUtils contains 69 programs, but we filter out 12 of them from our main results due to three reasons: (1) some utilities are trivial (e.g., `true`, `yes`) (2) certain utilities require special environments (e.g., `stty`) or privileges (e.g., `sudo`) not available in our experimental sandbox; and (3) certain utilities perform destructive operations (e.g., `rmdir`) that may break the testing harness.

only an initial set of seed tests, and does not iterate to add more tests. (2) *Coverage baseline:* a two-agent setup similar to ACToR, but the discriminator agent is only instructed to find additional tests that increase the line coverage of the C program, so the new tests do not specifically try to reveal inequivalent behavior. (3) *ACToR-NoFuzz:* ACToR but without the fuzzing tool. Comparing ACToR with the first two baselines demonstrates the benefit (if any) of our adversarial design. Comparing ACToR with ACToR-NoFuzz demonstrates the benefit (if any) of our proposed fuzzing tool.

**Evaluation Metrics.** Our primary evaluation metric to evaluate the correctness of an individual translation is *pass rate*, which is the percentage of tests that pass for a given set of tests. On our micro-benchmark, we report the pass rate on manually written validation tests when we evaluate different configurations of ACToR only (e.g., different LLM–agent settings or experiment hyperparameters). When we compare ACToR against other translation methods, we compute pass rates using a set of tests produced by a competing method, and refer to this metric as *relative pass rate*. This reduces bias from a method overfitting to its own tests (in practice, a method usually passes almost all of its own tests) and provides a stronger signal for cross-method comparison. We use relative pass rate for (i) ablation studies that compare ACToR against alternative designs on the micro-benchmark, and (ii) the final experiment on the macro-benchmark, which lacks manually written tests and therefore relies solely on test suites generated by the methods. The test sets used to compute relative pass rate achieve an average of 92.6% line coverage on the micro benchmark and 90.1% line coverage on the macro benchmark, by unioning test cases across translation methods. This indicates they are a useful signal for overall correctness as well.

**Qualitative Requirements.** ACToR aims to translate C code into fully safe Rust. Across both benchmarks, we enforced the absence of any `unsafe` blocks using a sanity check at every iteration. After all the experiments finished, we manually reviewed the final translations output by ACToR, confirming they are indeed safe and contain no malicious constructs intended to bypass the checking.

**Experimental Configuration.** We fixed a configuration to compare different agent settings and translation methods on the micro-benchmark. By default, ACToR and all baselines start with 15 manually crafted, diverse seed tests based on the C code for each program [3]. For the coverage baseline, ACToR-NoFuzz, and ACToR, we run 10 iterations of test generation and translation. On each iteration, we generate 3 new tests, resulting in a total of 45 tests per program in the end. We fix these parameters because exploring different configurations for all agent–model settings and translation methods would be prohibitively costly. We conduct an ablation experiment to study the influence of different experiment configurations on the final correctness and cost of ACToR. We apply the most cost-effective configuration to translate the macro-benchmark. The prompt template for the translator agent is shared among all the translation methods and put in Appendix A. The prompts for the test generator agent in the coverage baseline and the discriminator of ACToR are in Appendix A.

## 4.2 EVALUATION RESULT ON MICRO BENCHMARK

**Correctness & Adaptability.** We first test ACToR on the micro benchmark with different agents and LLMs. To analyze the correctness improvement, we compare ACToR approach with a *naive baseline* on pass rate. The experiment results per program are shown in Figure 2. All translations from ACToR and the naive baseline are 100% safe code and free of `unsafe` blocks. ACToR consistently improves the translation correctness on all 5 agent-model settings. With Claude Code and Claude-Sonnet-4.5, the validation pass rate rises from an average of 89.2% (one-shot naive baseline) to 98.2% after 10 iterations. For Mini-SWE-Agent, pass rates increase from 92.5% to 97.2% with Claude-Sonnet-4.5 and from 84.1% to 86.8% with GPT-5 mini on average. Across agent-LLM settings, the closed-source Claude Code, together with the Claude-Sonnet-4.5, achieves the highest final pass rate. We therefore use Claude Code + Claude-Sonnet-4.5 as the default configuration for the following experiments. When using Claude Code and Claude-Sonnet-4.5, it takes 411 million tokens to finish 6 programs, which costs $201 USD in total. The average time cost for ACToR to finish one program is approximately 2.45 hours[4].

---

[3]The seed tests carry intended test format that both translator and discriminator of ACToR use. It also avoids a cold-start that produces translations in overly poor quality at the first few iterations for ACToR as well as all other systems we compare to.

[4]The token consumption is computed by summing up the input, output, and cached tokens reported in the API response messages. When computing the time cost, we first compute the average time cost per iteration, then times it by 10 as an approximation of the total runtime for translating one program.

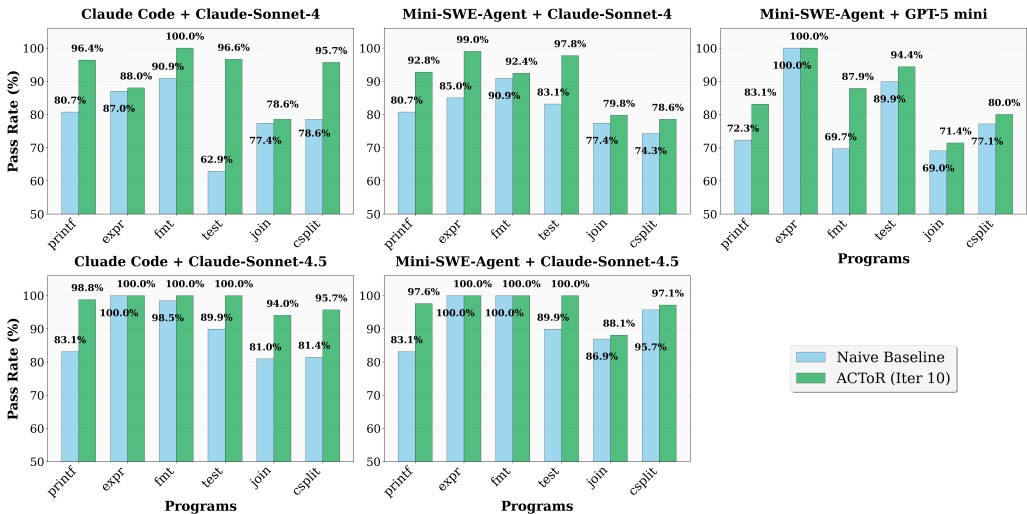

Figure 2: Overall correctness (pass rate) achieved by ACToR on micro benchmark across different settings compared with naive baseline, on 5 different agent-model settings.

**Ablation Study on Design.** We conduct an ablation to know how the adversarial design in ACToR drives correctness gains on the micro-benchmark. We analyze the improvement offered by two design choices: (a) having an *adversarial discriminator* in the workflow to guide the translator, and (b) forcing the discriminator agent to use the *fuzzing script* we provided.

We compare three variants of ACToR: coverage-baseline, ACToR-NoFuzz, and the full ACToR (with fuzzing). The results summed across programs at the final iteration are reported in Figure 3. Entry (row, column) shows the relative pass rate of the row method on the column method's tests. All three methods largely pass their own tests as expected. On coverage-baseline tests, ACToR-NoFuzz and the full ACToR reach 93% and 96% passing rates, respectively; in the reverse direction, coverage-baseline achieves less than 60% passing rate on tests from either ACToR variant. This indicates that adversarial discriminators produce tests that more effectively expose semantic mismatches and result in more correct translations. Between ACToR variants, the full ACToR attains 82% vs. 81% when cross-tested against ACToR-NoFuzz. This suggests that the discriminating agent uses the

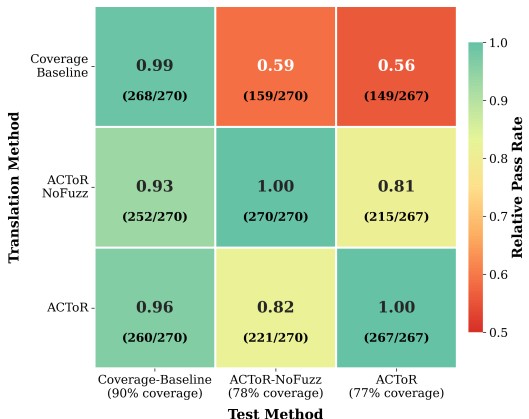

Figure 3: The relative comparison among 3 translation methods on Claude Code with Claude-Sonnet-4.5. Entry (row, column) is the relative pass rate of the row's translation on the column method's tests.

fuzzing script to surface more challenging bugs in the translated code. We also measure the final test coverage of each method on the C code. The result is shown at the bottom of Figure 3. The coverage baseline works as expected and achieves the highest line coverage on the C code than two other ACToR variants. However, it is worse in correctness. This further supports the conclusion that adversarial translation is more efficient in improving correctness compared with purely sampling new test cases based on coverage. Finally, we compare the financial costs of completing all six programs: the coverage baseline costs $95 USD, while ACToR-NoFuzz and the full ACToR cost $240 USD and $211 USD, respectively. To better understand cost-effectiveness, we ran an *equal-cost* experiment in which we extended the coverage baseline to 25 iterations, matching the approximate cost of running ACToR for 10 iterations. Even under equal budget, the coverage baseline does not close the performance gap: its relative pass rate (165/267 = 61.8%) remains far below that of ACToR

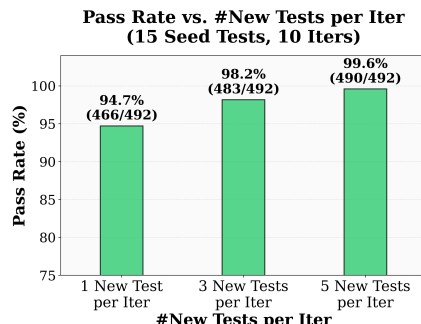 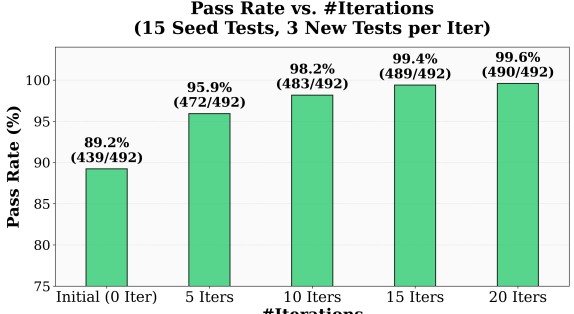

Figure 4: The validation pass rate of ACToR on different configurations. Pass rate versus the number of new test cases added per iteration (Left). Pass rate versus the number of iterations (Right).

(515/537 = 95.9%). This demonstrates that ACToR delivers substantially higher correctness per dollar spent. Additional details are shown in the Appendix B.

**Ablation Study on Configurations.** We conduct a second ablation study on the micro-benchmark to analyze how different experimental configurations affect ACToR 's translation performance. We examine three factors: *seed test cases*, *number of iterations*, and *number of test cases* generated by the discriminator agent per iteration. Recall that the default configuration of ACToR uses 15 initial seed tests, runs for 10 iterations, and adds 3 new tests per iteration. To assess the effect of the number of tests added per iteration, we run two variants of ACToR that follow the default setting except that they add 1 and 5 test(s) per iteration, respectively, instead of 3. To analyze the relation between correctness and the number of iterations, we run ACToR for 20 iterations with other parameters set as default. We collect the initial translation ($0^{th}$ iteration) and the translated code at the end of $5^{th}$, $10^{th}$, $15^{th}$, and $20^{th}$ iterations. We compute their *pass rates* on the validation test set. The results summed across programs are shown in Figure 4. Compared with adding 1 test per iteration, adding 3 new tests costs 16% more USD to translate all programs, and improves the final pass rate from 94.7% to 98.2%. Adding 5 new tests per iteration further increases the money cost by 29% compared with adding 3 tests per iteration, but only improves the pass rate by 1.4%. When comparing across different iteration numbers, the validation pass rate is improved by 9% after the first 10 iterations. Further extending to 20 iterations yields only a 1.4% improvement, at the expense of $2.1\times$ the total financial cost. At last, we analyze the influence of initial test cases. We run ACToR starting with only a single seed test, while keeping all other parameters as default values. A single seed test is sufficient to demonstrate the expected test case format and prevent trivial failures due to formatting mismatches, but it provides less initial coverage than the default setting of 15 seed tests. Using the default configuration (15 initial tests) increases cost by 6% relative to the single-seed setup, and improves the final pass rate from 94.9% to 98.2%. To balance performance and cost, we adopt the configuration of 15 seed tests, 10 iterations, and 3 new tests per iteration when translating the macro-benchmark.

**Stability of Results.** To assess the stability of ACToR 's translation results, we run ACToR three times using the default settings (i.e., 15 initial tests, 10 iterations, and 3 tests per iteration). We record the pass rate of each trial on the validation set. For each program, we compute the *sample standard deviation* of the three pass rates and then average these standard deviations across all programs. On average, the three runs achieve a pass rate of 97.0%, with a deviation of 1.9 percentage points.

> On the micro-benchmark, our system substantially and stably improves translation correctness compared with the naive baseline. The adversarial discriminator is crucial to improving correctness, and the fuzzing tool further strengthens the discovery of mismatches.

## 4.3 EVALUATION ON MACRO BENCHMARK

We compare ACToR results with the coverage baseline to examine the improvement achieved in correctness, using the *relative pass rate* metric. We use the Claude Code agent and Claude-Sonnet-4.5 for this experiment[5]. All the final translations output by ACToR are free of `unsafe` blocks.

---

[5]We run macro evaluation on both Claude-Sonnet-4 and Claude-Sonnet-4.5. Due to space limitations, we put the result of Claude-Sonnet-4 in Appendix C for reference. The results are consistent between two trials.

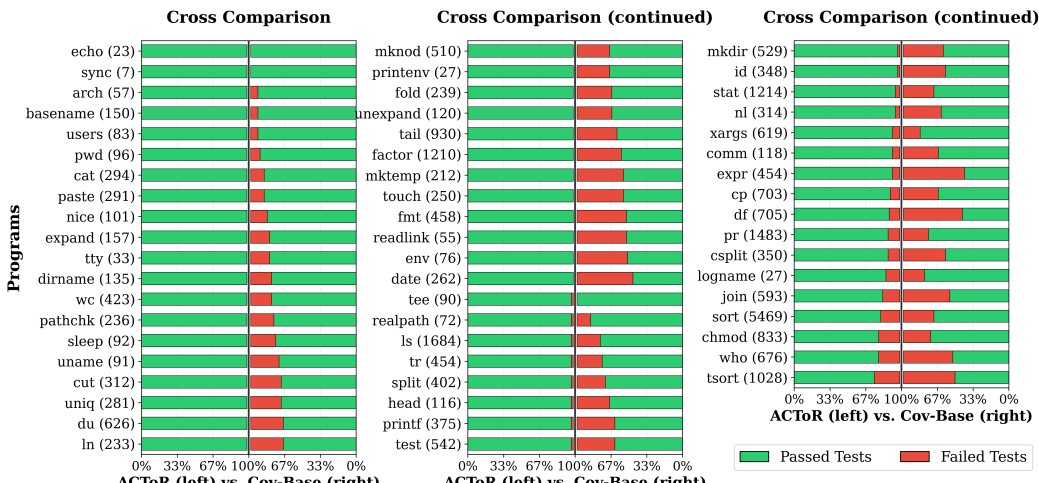

Figure 5: The relative pass rate when cross-comparing ACToR and coverage baseline (Cov-Base) on Claude Code with Claude-Sonnet-4.5 at iteration 10. For each program, the left bar shows evaluating the translation from ACToR on tests generated in coverage-baseline; the right bar is evaluating the translation from coverage-baseline on ACToR tests. The length of each program in LoC is presented next to the program name.

The cross-comparison result at the $10^{th}$ iteration is shown in Figure 10. The left bar for each program is the relative pass rate of testing ACToR's translation on the test cases from the coverage baseline. The right bar shows the reverse: relative pass rate of the coverage baseline against ACToR's test cases. We see that ACToR outperforms the coverage baseline on 54/57 programs on the relative pass rate. ACToR gets 100% relative pass rate on 32 programs when running on the tests from the coverage baseline, while the coverage baseline can only get full pass on 3 programs. On 57 programs, ACToR achieves an average 96.4% relative pass rate, 25.1% higher than the 71.3% relative pass rate obtained by the coverage baseline. The cross-comparison confirms that adversarial agentic design in ACToR plays a significant role in improving the translation correctness. In an absolute (non-relative) sense of correctness, by taking the union of the test cases from both ACToR and the coverage baseline, we get a test set of an average 90.1% coverage. ACToR achieves a 95.9% pass rate on it. This demonstrates the utility of ACToR on translating real-world projects with high correctness. The total financial costs to translate 57 programs using coverage-baseline and ACToR are $808 USD and $1634 USD, respectively. On average, ACToR consumes 57 million tokens per program, with more than 97% of them being cache-read tokens. The average time cost for ACToR to finish one program is about 2.5 hours.

> ACToR automatically translates the BSDCoreUtil benchmark with high correctness with reasonable costs, outperforming the coverage baseline on 54/57 programs and achieving an average of 96.4% relative pass rate.

## 5 CONCLUSION

We present ACToR, an agent-based automatic translation approach that aims for functionally correct C to Rust translations. Our method introduces a collaborative pipeline in which two agents iteratively refine the translations. At each iteration, the discriminant agents detect the semantic mismatches between the source code and the translated program, and report them as failing test cases. The translator agents then improve the translation to pass the current test set. Our experiments demonstrate the power of ACToR's adversarial agentic setup to produce translations with high correctness, surpassing the quality of translation as compared to non-adversarial setups. A noteworthy advantage that ACToR brings to the translation landscape is the potential for translating a large number of command-line programs with minimal or no human intervention, making ACToR the first approach of this kind.

## REPRODUCIBILITY STATEMENT

The full implementation of ACToR, along with detailed instructions, benchmark programs, and evaluation scripts, is provided in the anonymized supplementary artifact. Section 3 describes the algorithmic design of ACToR, including assumptions and iteration parameters, while Section 4 outlines our evaluation methodology and benchmark datasets. Appendix B includes the prompt template used for the translator and discriminator agents. Together, these resources enable independent verification of both our experimental setup and results.

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

```
(ACToR/ Coverage Baseline) Translator Agent Prompt

Task Description:
```
You are an expert in C and Rust.
Your task is to translate the C `<project_name>` project to safe Rust implementations.

<...Project Setup>

## Workflow
1. Read the C code and the test script to understand the functionalities.
2. Initialize a new Cargo project in the `ts/` folder.
3. Translate the C code to Rust code and compile it into binary.
4. Run `./testcmp.sh` to compare the output of the translated Rust code with the
     original C program. You should run `./testcmp.sh --help` to understand how to use
     the test script.
5. Clean the working directory by removing temporary files and backup files.

## constraints
- The translated Rust code MUST compile and MUST be 100% safe.
- The translated Rust code MUST pass all the unit tests.
```
```

Figure 6: The task prompt for the translator agent.

```
(Coverage Baseline) Test Generator Agent Prompt — Adding Tests

Task Description:
```
You are an expert in C and Rust.
Your task is to add additional test cases to for the C program to improve the coverage.

<...Project Setup>

## Workflow
1. Read the C code to understand the functionalities.
2. Focus first on **core functionalities**, then explore **edge cases**.
3. Run `make clean && make all && ./test.sh coverage` to compile the C code and get the
     current coverage.
4. Read the coverage report and the record of added test cases in `test_cases_record.md`
      to find potential missed cases.
5. Design **3** new test cases that are different from existing test cases.
6. Run `./test.sh coverage` to get the new coverage. Ensure that the new coverage is
     higher than the previous one.
7. Clean the working directory by removing temporary files and scripts, temporary test
     cases, and backup files.

<...Test formatting requirements>

## Constraints
1. There should be exactly 3 new test cases added to the JSONL file. You should run `./
     testcmp.sh` and the number of test cases will be shown. There should be `<
     current_test_cases_number> + 3` test cases in total.
2. The 3 test cases should be different from each other. You should check this by
     reading the content of the test cases.
3. The added tests must be valid for the C code. You should run `make clean && make all`
      and then run `./testcmp.sh compare ./xxx.out(compiled from C code)`. It must show `
     All tests passed!`.
```
```

Figure 7: The task prompt for the test generator agent of the coversge baseline.

## A    PROMPT TEMPLATE

For Mini-SWE-Agent, we keep the system prompt that describes the agent workflow unchanged
as the original one during all the experiments. For Claude Code, we begin directly with the task
described in the user message, without any additional text preceding it. The task prompt structure for
the translator agent is shown in Figure 6. This prompt is shared among all translation methods during
evaluation. The prompt structure for the test-generation agent used in the coverage baseline is shown

```
(ACToR) Discriminator Agent Prompt — Adding Tests

Task Description:
```
You are an expert in C and Rust.
Your task is to add additional test cases to discover semantic mismatches between the C
    code and the translated Rust code.

<...Project Setup>

## Workflow
1. Analyze the C code and the translated Rust code to detect **semantic mismatches**.
2. Focus first on **core functionalities**, then explore **edge cases**.
3. Read the current test script and the record of added test cases in 'test_cases_record
    .md' to find potential missed cases.
4. Collect the best **3** new input cases that expose mismatches between C code and Rust
    translation. Add the 3 new test cases to the test cases file.
5. Run the new tests to compare the output of the translated Rust code with the original
    C program to confirm the mismatches.
6. Clean the working directory by removing temporary files and scripts, temporary test
    cases, and backup files.

<...Test formatting requirements>
IF {$allow_fuzz}
<...Details on the usage of the fuzzer script>
ENDIF

## Constraints
1. There should be exactly 3 new test cases added to the JSONL file. You should run './
    testcmp.sh' and the number of test cases will be shown. There should be '<
    current_test_cases_number> + 3' test cases in total.
2. The 3 test cases should be different from each other. You should check this by
    reading the content of the test cases.
3. The added tests must be valid for the C code. You should run 'make clean && make all'
    and then run './testcmp.sh compare ./xxx.out(compiled from C code)'. It must show '
    All tests passed!'.
4. The added tests should reflect the differences between the C code and the Rust code.
    You should run './testcmp.sh compare ./ts/target/release/xxx(compiled from Rust code
    )'. The Rust code should fail on all 3 new test cases.
```
```

Figure 8: The task prompt for the discriminator agent of ACToR.

```
Post-processing Prompt — Eliminating Unsafe Blocks

Task Description:
```
You are an expert in C and Rust.
Your task is to translate the C '<project_name>' project to safe Rust implementations.

<...Project Setup>

## Workflow
1. Read the C code and the test script to understand the functionalities.
2. Initialize a new Cargo project in the 'ts/' folder.
3. Translate the C code to Rust code and compile it into binary.
4. Run './testcmp.sh' to compare the output of the translated Rust code with the
    original C program. You should run './testcmp.sh --help' to understand how to use
    the test script.
5. Clean the working directory by removing temporary files and backup files.
```

The translation contains unsafe keywords.
Please fix it to ensure no unsafe code is used.
Also, keep it passing all the tests.
```

Figure 9: The task prompt for eliminating unsafe blocks.

in Figure 7. The prompt structure for the discriminator agent for ACToR is shown in Figure 8. The fuzzing mode is controlled by the 'allow_fuzz' section in the prompt.

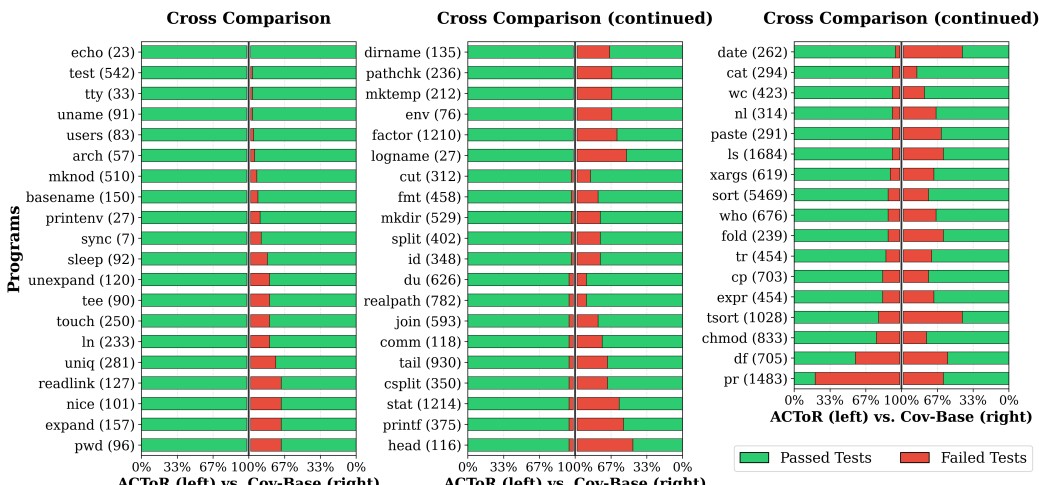

Figure 10: The relative pass rate when cross-comparing ACToR and coverage baseline (Cov-Base) on Claude Code with Claude-Sonnet-4 at iteration 10. For each program, the left bar shows evaluating the translation from ACToR on tests generated in coverage-baseline; the right bar is evaluating the translation from coverage-baseline on ACToR tests. The length of each program in LoC is presented next to the program name.

## B  EQUAL-COST COMPARISON ON MICRO BENCHMARK

Running all six programs in the micro benchmark costs $95 USD for the coverage baseline and $211 USD for ACToR. To better understand cost-effectiveness, we ran an *equal-cost* experiment in which we extended the coverage baseline to 25 iterations to match the cost of running ACToR for 10 iterations. We cross-compare the resulting translations using *relative pass rate*. The two methods complete different numbers of iterations, resulting in different numbers of test cases. The total USD cost for the coverage baseline to finish 25 iterations is $220 USD, roughly the same as ACToR to finish 10 iterations ($211 USD). However, even under equal budget, the coverage baseline does not achieve comparable correctness. Its relative pass rate (165/267 = 61.8%) remains far below that of ACToR (515/537 = 95.9%). In absolute terms, ACToR fails only 22 out of 537 tests, while the coverage baseline fails 102 out of 267. This demonstrates that ACToR delivers substantially higher correctness per dollar spent. Additional details are shown in the appendix.

## C  MACRO EXPERIMENTS WITH CLAUDE CODE AND CLAUDE-SONNET-4

We also run ACToR with Claude-Sonnet-4 to translate the macro-benchmark. In this trial, we use a slightly different implementation that discourages the use of unsafe, but does not enforce full safe Rust in the middle of the translation loop. We enforce full safe Rust at the end of the translation, as a post-processing step. This post-processing is performed via a single agent call that is required to eliminate all `unsafe` blocks while preserving the functionality of translated Rust code. The prompt for this post-processing step is shown in Figure 9. We emphasize that this strategy slightly differs from the approach used in the main paper when run ACToR with Claude-Sonnet-4.5, where we enforce full safety at *every* iteration of the translation loop. All other experimental settings in this trial are identical to those described in the main text. The results are presented in Figure 10. With Claude-Sonnet-4, ACToR outperforms the coverage baseline on 55/57 programs in terms of relative pass rate. ACToR achieves full pass on 26 programs using the tests from the coverage baseline. Across all 57 programs, ACToR attains an average relative pass rate of 93.9%, which is 18.9% higher than the 75.0% pass rate of the coverage baseline. ACToR achieves a 95.3% pass rate on the union of the test cases from both ACToR and the coverage baseline. This result aligns with the findings on Claude-Sonnet-4.5, demonstrating that ACToR's performance is stable and generalizes across different model configurations.

## D    THE USE OF LARGE LANGUAGE MODELS (LLMs)

Large Language Models (LLMs) were used exclusively as a writing aid. Their role was limited to polishing grammar. LLMs did not contribute to research innovation, design, or analysis. All substantive content, methodology, and conclusions are the sole work of the human authors.

