# OpenReview forum: "Adversarial Agent Collaboration for C to Rust Translation"
_ICLR.cc/2026/Conference — Submitted to ICLR 2026_

### Official Review · Reviewer_NQvr · 2025-10-27

**Soundness:** 2
**Presentation:** 3
**Contribution:** 3
**Rating:** 4
**Confidence:** 4

**Summary:**

This paper proposes ACToR (Adversarial C To Rust translator), an adversarial two-agent workflow for translating C programs to safe Rust. A translator LLM agent iteratively produces a Rust version, while a discriminator agent searches for counterexample tests that expose behavioral mismatches. The newly found failing tests are added to the test suite. The method is evaluated on two benchmarks: a micro-benchmark and a macro-benchmark. ACToR improves translation correctness by up to 18.9% compared to the baseline approach.

**Strengths:**

1. Novel and Effective Framework: This work tries to address a well-known weakness in automated code generation, i.e., the difficulty of ensuring semantic equivalence beyond a fixed set of tests
2. Good ablation studies: the paper isolates the effect of adversarial test generation and a simple fuzzer
3. Include evaluation with the agentic frameworks.

**Weaknesses:**

1. Correctness is proxied by tests. However, tests may not be able to expose real bugs. No human validation of the translated code or the quality / sufficiency of the test cases.
2. I am wondering how well this method could generalize to other existing datasets, e.g., CRUST-Bench: A Comprehensive Benchmark for C-to-safeRust Transpilation?
3. The cost estimation (e.g., tokens / API cost) seems to be neglected for comparison and evaluation of different approaches.
4. There are many papers that use LLMs for code translation. I am wondering about the technical novelty compared with prior approaches. Also, other papers claim that they do repo-level translation. Why do you claim that your work is the first to work on large-scale programs? Is it really large-scale compared with repo-level's prior work?

**Questions:**

1. How do you compare your benchmark to "CRUST-Bench: A Comprehensive Benchmark for C-to-safeRust Transpilation"?
2. Is there any guarantee that the final generated Rust programs are really memory-safe? Or the agents could also escape certain checks to bypass? How much confidence in correctness can you get out of the generated Rust programs? Is there any manual validation?
3. What if you only use the fuzzing script without the agent?
4. Can you compare your approach with the other paper, "Exploring and Unleashing the Power of Large Language Models in Automated Code Translation", and tell me what your technical novelty is? Similarly, for this paper, "AlphaTrans: A Neuro-Symbolic Compositional Approach for Repository-Level Code Translation and Validation".

---

> ### Author Response · Authors · 2025-11-22
> **Author Response (Common Response to All Reviewers)**
>
> We thank the reviewer for the useful and constructive comments. As a recap, ACToR’s main technical contribution is to show that GAN-style architecture improves over architectures that use a single LLM agent in C to safe Rust translation, keeping all other variables (type of benchmarks, test environment, etc.) constant.
>
> Based on the suggestions from all the reviewers, **we have made the following improvements to the paper**, wherein:
>
> - We have qualified the main claim, restricting it to standalone command-line utilities only; this matches our reported benchmarks precisely. We clarified that our problem statement has been to generate fully safe Rust in Section 3, and stated the quality requirement in Section 4.1.
>
> - We added a new ablation study to explore different configurations of ACToR, including: different numbers of iterations and new tests per iteration, and different seed test quantities. Our main claims and conclusions continue to hold across these ablation experiments.
>
> - We added a repeatability experiment to assess the stability of ACToR on the micro benchmark. Across 3 trials, ACToR shows a low standard deviation on test pass rate.
>
> - We reran ACToR on the macro benchmark, with a strict safety requirement at each iteration, and removed the post-processing step. The results are consistent with the existing one: All Rust codes produced by ACToR are fully safe, and exhibit similar improvement in correctness compared with baseline. We moved the previous macro result to the Appendix.
>
> - We reported details of financial, token, and runtime costs for both micro- and macro-experiments. We added an equal-cost comparison to demonstrate the cost-effectiveness of ACToR.
>
> We use Claude Code with Claude-Sonnet-4.5 for the new and rerun experiments. Sonnet-4.5 has an identical cost as Sonnet-4 and completed the experiments noticeably faster in practice, enabling us to finish the additional experiments within the rebuttal timeframe. Importantly, our main claims and conclusions hold under Sonnet-4.5. The new results are consistent with those obtained using Sonnet-4.
>
> We have uploaded the revised paper and included a version highlighting the updates in the supplementary material.
>
> The answers to specific questions raised are below.

---

> ### Author Response · Authors · 2025-11-22
> **Author Response (Answer to Weaknesses and Questions 1/3)**
>
> #### **W1. Correctness is proxied by tests. However, tests may not be able to expose real bugs. No human validation of the translated code or the quality / sufficiency of the test cases.**
>
> We agree that approximate correctness vs. exact semantic equivalence is a continuum which can be explored with higher tests. However, approximate correctness based on tests is a practical choice, since formal specifications of C behavior are difficult to obtain for most real-world programs. In practice, code-migration efforts rely heavily on unit tests—for example, in the DARPA program [1] and in the Rust reimplementation of GNU Coreutils [2].
>
> We understand your concerns about the quality of test cases generated by agents. A useful indication is that the translated code from the coverage baseline (also an iterative agentic translator) failed 29%~40% of the tests generated by ACToR. The original C code can pass all of them. This suggests that, from the translation perspective, the test generated by ACToR is sufficiently discriminative to expose translation errors.
>
> ---
>
> #### **W2. How does ACToR generalize to other existing datasets, e.g., CRUST-Bench?**
>
> We believe that our main claim: GAN-style iteration is more beneficial than single LLM architecture likely holds beyond our specific benchmarks. We have, however, qualified our claims in the revised draft to be restricted to standalone Unix utilities now, since our evaluation is on that.
>
> The CRUST-Bench exclusively has library code, whereas our evaluation has been on standalone programs which are testable independently. Library API and their signatures can change heavily when they are migrated from C to Rust and change based on the choice of test cases. One needs to have tests that capture all intended ways (and sequences) in which a library’s multiple APIs can be called. A recent work highlights tricky API precondition requirements when constructing test cases for mapped APIs across languages [3]. The CRUST-Bench, on the other hand, only has fixed test driver programs manually constructed for each library in the current form.
>
>
> Because modeling correct translation of libraries adds its own additional layer of complexity, it distracts from the central point of our paper. Our main goal has been to measure whether a single agent vs. our GAN-style architecture is better for the task. To keep the orthogonal issue of test-suite completeness out of the scope, we chose to work with standalone programs as their external interface remains unchanged between C and Rust.
>
> In our revised paper, we have explicitly qualified our claims to be limited to standalone Unix utilities. Notwithstanding, we would like to point out that our standalone programs are largely of comparable size to the majority of programs in CRUST-Bench.
>
>
> ---
>
> #### **W3. The cost estimation (e.g., tokens / API cost) seems to be neglected for comparison and evaluation of different approaches.**
>
> We added the token and financial cost analysis for both micro and macro experiments in the revised paper (line 370 and line 518). We include financial cost comparison across different translation approaches in the ablation study and in the final evaluation on the macro benchmark.
>
> On both micro and macro benchmarks, ACToR costs roughly twice as much as the coverage baseline. Despite the higher expense, ACToR delivers a substantial improvement in correctness, raising the relative pass rate from 56% to 96% on the micro benchmark and from 71% to 96% on the macro benchmark.
>
> To better understand cost-effectiveness, we add an equal-cost experiment  (lines 426~447 and Appendix B) on the micro-benchmark, in which we extended the coverage baseline to **25** iterations (220 USD in total), matching the cost of running ACToR for 10 iterations (211 USD). We cross-compare the resulting translations using relative pass rate (i.e., using the tests from coverage baseline to test ACToR, vice versa). Note that the two methods complete different numbers of iterations, resulting in different numbers of test cases. Even under the same budget, the coverage baseline does not achieve comparable correctness. Its relative pass rate (165/267 = **61.8%**) **remains far below that of ACToR** (515/537 = **95.9%**). ACToR fails only 22 out of 537 tests from the coverage baseline, while the coverage baseline fails 102 out of 267. This demonstrates that ACToR delivers substantial improvement in translation correctness under a fixed budget.
>
> ---
>
> [1] DARPA. Tractor: Translating all c to rust. DARPA website, 2025. URL https://www.darpa.mil/research/programs/translating-all-c-to-rust.
>
> [2] uutils/coreutils. Cross-platform Rust rewrite of the GNU coreutils website. 2025. URL https://github.com/uutils/coreutils
>
> [3] Ibrahimzada, A.R., Paulsen, B., Jabbarvand, R., Dodds, J. and Kroening, D., 2025. MatchFixAgent: Language-Agnostic Autonomous Repository-Level Code Translation Validation and Repair. arXiv preprint arXiv:2509.16187.

---

> ### Author Response · Authors · 2025-11-22
> **Author Response (Answer to Weaknesses and Questions 2/3)**
>
> #### **W4. I am wondering about the technical novelty compared with prior approaches. Also, other papers claim that they do repo-level translation. Why do you claim that your work is the first to work on large-scale programs?**
>
> The main novelty is in the GAN-style framework to improve functional correctness of the translation and generalize to unseen tests. While being simple in hindsight, we have shown that ACToR is the first system that needs no human intervention to translate Unix system utilities with a total of ~30k LoC reliably (over 90% test passing rate) to safe Rust code. In our revised draft, we have made this claim clearer.
>
> Existing repo-level approaches do not provide sufficient automation to complete whole-program translations. This is because these systems are built on vanilla LLMs with fixed workflow, which are brittle in out-of-design cases and thus lack full automation. So they may introduce human intervention into the system [7,8,10], adjust their goal to fragment-correctness or compilation [6,9], allow unsafe Rust [4,10], or struggle with long programs [5].
>
> ---
>
> #### **Q1. How do you compare your benchmark to CRUST-Bench?**
>
> Continuing from weakness 2, our benchmark and CRUST-Bench target different program types. Our benchmark includes 63 popular Unix utilities, such as "ls" and "cat", each with an expressive test harness that supports adding additional tests easily. CRUST-Bench consists of 100 library programs collected from GitHub, with fixed Rust interfaces and test drivers.
>
> ---
>
> #### **Q2. Is there any guarantee that the final generated Rust programs are really memory-safe? Or the agents could also escape certain checks to bypass? How much confidence in correctness can you get out of the generated Rust programs? Is there any manual validation?**
>
> Concerns about the safety and correctness of agent-generated Rust code are understandable. We address these points below.
>
> **Safety.** We enforce a strict safety check at every iteration in all experiments. This ensures that any intermediate Rust program containing "unsafe" blocks or violating the safety constraints is immediately rejected and regenerated. After all experiments were complete, we manually inspected the ACToR’s outputs and confirmed that the generated programs contain no constructs intended to bypass safety checks. We added a dedicated paragraph in Section 4.1 that states these qualitative requirements to make the expectations and guarantees explicit.
>
> **Correctness.** We explicitly instruct the translator agent to faithfully implement the full functionality of the source C program without altering or simply semantics. We provide multiple lines of evidence supporting the translation quality:
> - In the micro benchmark, where we manually crafted test cases that cover diverse functionalities and achieve high line coverage (89%), ACToR achieved a high pass rate (98%).
> - In the macro benchmark where we apply cross-validation, the ACToR’s test cases are sufficiently discriminating at discovering translation bugs, as the coverage baseline failed many of them (29%). The codes produced by ACToR have a high pass rate (95%) on these discriminating test cases, indicating their correctness.
> - To further measure the quality of translation, we randomly sampled 10 programs ('wc', 'ls', 'tsort', 'mknod', 'cat', 'factor', 'pr', 'pathchk', 'mkdir', 'head') from the macro benchmark and manually inspected the preservation of main functionality, using the syntactic structure mapping as a reference. We found no obvious functional truncation or overfitting.
>
> ---
>
> [4] Eniser, H.F., Zhang, H., David, C., Wang, M., Christakis, M., Paulsen, B., Dodds, J. and Kroening, D., 2024. Towards translating real-world code with llms: A study of translating to rust. arXiv preprint arXiv:2405.11514.
>
> [5] Yang, A.Z., Takashima, Y., Paulsen, B., Dodds, J. and Kroening, D., 2024. Vert: Verified equivalent rust transpilation with few-shot learning. arXiv preprint arXiv:2404.18852, 26.
>
> [6] Shiraishi, M. and Shinagawa, T., 2024. Context-aware code segmentation for c-to-rust translation using large language models. arXiv preprint arXiv:2409.10506.
>
> [7] Shetty, M., Jain, N., Godbole, A., Seshia, S.A. and Sen, K., 2024. Syzygy: Dual code-test c to (safe) rust translation using llms and dynamic analysis. arXiv preprint arXiv:2412.14234.
>
> [8] Zhou, T., Lin, H., Jha, S., Christodorescu, M., Levchenko, K. and Chandrasekaran, V., 2025. LLM-Driven Multi-step Translation from C to Rust using Static Analysis. arXiv preprint arXiv:2503.12511.
>
> [9] Ou, G., Liu, M., Chen, Y., Du, X., Wang, S., Zhang, Z., Peng, X. and Zheng, Z., 2025. Enhancing llm-based code translation in repository context via triple knowledge-augmented. arXiv preprint arXiv:2503.18305.
>
> [10] Bai, Y. and Palit, T., 2025. RustAssure: Differential Symbolic Testing for LLM-Transpiled C-to-Rust Code. arXiv preprint arXiv:2510.07604.

---

> ### Author Response · Authors · 2025-11-22
> **Author Response (Answer to Weaknesses and Questions 3/3)**
>
> #### **Q3. What if you only use the fuzzing script without the agent?**
>
> To clarify, the provided fuzzing script is intended to assist the agent, not to replace it. General-purpose fuzzers (such as AFL) or our simple script alone cannot achieve comparable performance. This limitation stems from a well-known challenge [1] in fuzzing programs with diverse option sets—such as the system-utility programs in our benchmarks—where the reachable code paths depend on specific command-line options, input data with strict grammars, or even the execution environment. Effective fuzzing therefore requires customization around the functionality of each target program.
>
> In our setup, the discriminator agent uses the simple, agent-friendly fuzzing script as a starting point, but then adapts the script, prepares files, and configures the environment based on its understanding of the C code. This allows the agent to efficiently discover subtle tests that standalone fuzzers or the script alone would be unable to generate.
>
> ---
>
> #### **Q4. Can you compare your approach with the other paper, "Exploring and Unleashing the Power of Large Language Models in Automated Code Translation", and tell me what your technical novelty is? Similarly, for this paper, "AlphaTrans: A Neuro-Symbolic Compositional Approach for Repository-Level Code Translation and Validation".**
>
> These two papers’ technical choice is built on top of vanilla LLMs, constraining LLMs in strict workflows. In practice, their automation level is limited in terms of producing a runnable whole program translation. Thus, their research goal is also limited to pass the bounded tests/code fragment correctness, i.e., compilable function etc, but still far from functioning whole program translation.
>
> By contrast, we are building on top of recent autonomous coding agents, which do not need to manually implement workflows that are typically fragile. Coding agents have a much higher automation level. Therefore, our research goals focus on whole program correctness, especially the generalization to unseen test cases.
>
> Autonomous agents are promising, but directly giving them the task does not yield satisfactory results, as shown in the paper (the naive baseline and coverage baseline). Therefore, we introduce ACToR, which exploits multiple rounds of interaction between the translator and discriminator. ACToR is designed to push the translated code to generalize beyond the fixed test suite. In our evaluation, ACToR translation achieves high correctness when measured on unseen test cases.
>
> As a summary, two designs contribute to ACToR’s full automated handling of 57 command-line utility programs that prior work does not exhibit. One is leveraging recent advancements in autonomous coding agents, and the second is using a translator-discriminator framework to address coding agents' limitations in correctness.
>
> The technical ideas of these two papers, such as test-case decomposition or per-fragment validation, could be incorporated into ACToR to further improve the discriminator and translator agents in future work.
>
> ---
>
> [11] Zhang, Z., Klees, G., Wang, E., Hicks, M. and Wei, S., 2023. Fuzzing configurations of program options. ACM Transactions on Software Engineering and Methodology, 32(2), pp.1-21.

---

### Official Review · Reviewer_hYZH · 2025-10-30

**Soundness:** 2
**Presentation:** 3
**Contribution:** 3
**Rating:** 6
**Confidence:** 4

**Summary:**

This paper tackles the task of translating C code to safe Rust code with an adversarial setup of LLM agents. By having an agent that performs the translation interact with another that comes up with tests that expose the mismatch between C and Rust code, ACToR steers translation towards general correctness instead of overfitting to a fixed test suite. The authors showed that ACToR is effective on 63 real-world programs of non-trivial size and outperforms naive, coverage-guided, and ablated approaches.

**Strengths:**

This work addresses an important problem using an effective technique that makes intuitive sense and is well-executed. It makes sense to grow the test suite dynamically, and in addition for tests to maximally distinguish code in the two languages. Giving the discriminator agent access to a fuzzing script is a nice touch. As the authors convincingly demonstrate through pass rates relative to each competing tools, these bits give ACToR the edge over baselines and ablations. The paper is well-written, well-structured, and clearly presents this approach and findings. While not ground-breaking and despite a few issues below, I find this work to be a meaningful contribution to the field overall.

**Weaknesses:**

My complaints with the paper are mostly the following:

1. Unsound problem definition. The authors' definition of C-to-Rust translation does not require exclusive use of safe features, yet claims that "memory safe handling of malicious inputs is guaranteed by the Rust compiler itself." Since unsafe code may be involved, this assumption does not hold - it is entirely possible for inputs outside the valid universe U to expose vulnerabilities on the Rust side, which may be the same as or different from those on the C side, and which, at worst, are new vulnerabilities altogether. As such scenarios are not considered in the behavior equivalence check, such a problem definition (and any of its solutions) would defeat the purpose of what motivates translating from C to Rust in the first place (though ACToR apparently does not literally address this problem definition, as it is said to enforce safe Rust as a post-processing step).

2. Unprincipled experimental configuration. The decision to use 10 turns and 3 tests each turn feels handwavy. It seems entirely likely that with more turns and more tests added, more divergences between the C code and Rust code would be exposed. While I understand that the authors are limited by the cost of running the LLMs, I think it is very worthwhile to experimentally study the trade-offs between various (# turns, # tests) configurations and how they impact the degree of translation correctness ACToR can achieve.

**Questions:**

1. Could the authors comment on the issue with the unsound problem definition?

    Relatedly, what does the authors mean by a post-processing step that enforces safe Rust? Is it a check that looks for unsafe code, or does it transform unsafe code to safe code? Regardless, it is my opinion that a problem definition that requires safe Rust use and a design that enforces safety in transit would make ACToR more principled and impactful. One benefit, for instance, is that keeping Rust code safe throughout the process would prevent overhead associated with backtracking from unsafe code and steer the translation along a optimistic/promising direction. Modifying ACToR in this respect should not be too difficult to implement.

2. Would the authors consider studying the trade-offs between various experimenal configurations and their impact on translation correctness?

Minor: Could the authors briefly comment on how the "15 manually crafted, diverse seed tests" are created? I would also be curious to learn how this initial test suite could impact ACToR's performance. Would the authors be open to consider doing an ablation study on this?

---

> ### Author Response · Authors · 2025-11-22
> **Author Response (Common Response to All Reviewers)**
>
> We would like to thank the reviewer for the useful and constructive comments. As a recap, ACToR’s main technical contribution is to show that GAN-style architecture improves over architectures that use a single LLM agent in C to safe Rust translation, keeping all other variables (type of benchmarks, test environment, etc.) constant.
>
> Based on the suggestions from all the reviewers, **we have made the following improvements to the paper**, wherein:
>
> - We have qualified the main claim, restricting it to standalone command-line utilities only; this matches our reported benchmarks precisely. We clarified that our problem statement has been to generate fully safe Rust in Section 3, and stated the quality requirement in Section 4.1.
>
> - We added a new ablation study to explore different configurations of ACToR, including: different numbers of iterations and new tests per iteration, and different seed test quantities. Our main claims and conclusions continue to hold across these ablation experiments.
>
> - We added a repeatability experiment to assess the stability of ACToR on the micro benchmark. Across 3 trials, ACToR shows a low standard deviation on test pass rate.
>
> - We reran ACToR on the macro benchmark, with a strict safety requirement at each iteration, and removed the post-processing step. The results are consistent with the existing one: All Rust codes produced by ACToR are fully safe, and exhibit similar improvement in correctness compared with baseline. We moved the previous macro result to the Appendix.
>
> - We reported details of financial, token, and runtime costs for both micro- and macro-experiments. We added an equal-cost comparison to demonstrate the cost-effectiveness of ACToR.
>
> We use Claude Code with Claude-Sonnet-4.5 for the new and rerun experiments. Sonnet-4.5 has an identical cost as Sonnet-4 and completed the experiments noticeably faster in practice, enabling us to finish the additional experiments within the rebuttal timeframe. Importantly, our main claims and conclusions hold under Sonnet-4.5. The new results are consistent with those obtained using Sonnet-4.
>
> We have uploaded the revised paper and included a version highlighting the updates in the supplementary material.
>
> The answers to specific questions raised are below.

---

> ### Author Response · Authors · 2025-11-22
> **Author Response (Answer to Weaknesses and Questions)**
>
> #### **W1 (Q1). Unsound problem definition. The authors' definition of C-to-Rust translation does not require exclusive use of safe features.**
>
> > Could the authors comment on the issue with the unsound problem definition?
>
> Our goal has always been to generate pure safe Rust only, in addition to the correctness mentioned. We had forgotten to say this explicitly in Sec 3.1, but as reported in our evaluation, we had made sure that all Rust programs are completely safe Rust. This was done using a post-processing pass that checks and eliminates the “unsafe” blocks.
>
> In our updated draft, we have revised the definition in Sec 3.1 and have taken your suggestion to instruct the agents to **enforce safe Rust in each iteration**. If not, we repeat the steps in each iteration until the output is safe Rust. We have rerun the macro experiment using this new setting. The end result retains the overall same high pass rate, and has the same or higher correctness improvement compared with the coverage baseline. All final programs are **fully safe Rust**.
>
> > Relatedly, what does the authors mean by a post-processing step that enforces safe Rust?
>
> In the original evaluation, we relaxed the unsafe check during iterations; after all iterations were finished, we added another simple agent call to remove all unsafe code. We instruct the agent to eliminate unsafe blocks while keeping the functionality. The details are provided in Appendix C of the revised paper.
>
> In our revised draft, we have changed this. All the new experiments enforce a strict safety check after each iteration. Therefore, the post-processing step is no longer needed in new experiments.
>
> ---
>
> #### **W2 (Q2). Unprincipled experimental configuration. The decision to use 10 turns and 3 tests each turn feels handwavy.**
>
> To address this, we have reported on an additional ablation study on the experimental configurations (lines 448~469) in the revised draft. We analyzed the influence of the number of iterations and the number of added tests per iteration on the final pass rate and cost. We find that **10 iterations and 3 tests per iteration** are nearly the most cost-efficient trade-off, and use this configuration in the macro evaluation. We have reported the time, financial cost, and tokens to support the choice of parameters.
>
> ---
>
> #### **Q3. Could the authors briefly comment on how the "15 manually crafted, diverse seed tests" are created? I would also be curious to learn how this initial test suite could impact ACToR's performance. Would the authors be open to considering doing an ablation study on this?**
>
> We crafted the seed tests based on the C code and its manual to cover diverse functionality and to illustrate different testing formats. For system utility programs, their seed tests aim to demonstrate multiple command-line options and test file preparation, etc. This diversity also sets a correctness standard for the early iterations, helping ACToR avoid producing overly low-quality translations at the start. Note that the same seed tests are used for ACToR and other methods.
>
> To study the impact of diverse initial tests, we have added an experiment in the revised draft with **1** initial test (lines 462~468). We found that using 15 seed tests **improves the final pass rate from 94.9% to 98.2%** compared to starting with a single test. Therefore, we use 15 seed tests as the configuration for translating the macro benchmark.

---

### Official Review · Reviewer_vkya · 2025-11-01

**Soundness:** 2
**Presentation:** 3
**Contribution:** 3
**Rating:** 4
**Confidence:** 3

**Summary:**

The paper introduces ACToR, an adversarial two‑agent framework for translating C programs to safe Rust. A translator agent proposes a Rust program while a discriminator agent generates tests, including fuzzing‑aided cases, to expose behavioral mismatches, and the process iterates. On 6 micro programs and 57 BSDCoreUtils utilities (median 485 LOC), ACToR reports high test pass rates, outperforming non‑adversarial baselines and showing up to 18.9% relative pass‑rate gains on the macro benchmark.

**Strengths:**

- The paper addresses a timely and challenging problem: translating nontrivial C codebases to safe Rust with minimal human intervention.
- ACToR consistently improves over the naive baseline across three agent-model choices.

**Weaknesses:**

- The core comparisons are against a “naive” single-agent baseline and a “coverage” baseline. There is no head‑to‑head comparison with recent C→Rust systems that combine LLMs with analysis.
- The evaluation lacks runtime, token, and financial cost information, which is key for judging practicality at scale.
- Results appear to be single‑run without variance across seeds or model nondeterminism. Stability across runs is not reported. The text notes three programs that aborted before 10 iterations after multiple reruns, hinting at variability.
- The paper states “to our knowledge, it is the first such system that reliably translates C programs of this scale.” Given the breadth of recent efforts that blend LLMs with program analysis, it would be safer to qualify the claim with the particulars of the benchmark and evaluation protocol, or provide a stronger head‑to‑head against at least one recent approach on a shared subset.

**Questions:**

See Weaknesses.

---

> ### Author Response · Authors · 2025-11-22
> **Author Response (Common Response to All Reviewers)**
>
> We would like to thank the reviewer for the useful and constructive comments. As a recap, ACToR’s main technical contribution is to show that GAN-style architecture improves over architectures that use a single LLM agent in C to safe Rust translation, keeping all other variables (type of benchmarks, test environment, etc.) constant.
>
> Based on the suggestions from all the reviewers, **we have made the following improvements to the paper**, wherein:
>
> - We have qualified the main claim, restricting it to standalone command-line utilities only; this matches our reported benchmarks precisely. We clarified that our problem statement has been to generate fully safe Rust in Section 3, and stated the quality requirement in Section 4.1.
>
> - We added a new ablation study to explore different configurations of ACToR, including: different numbers of iterations and new tests per iteration, and different seed test quantities. Our main claims and conclusions continue to hold across these ablation experiments.
>
> - We added a repeatability experiment to assess the stability of ACToR on the micro benchmark. Across 3 trials, ACToR shows a low standard deviation on test pass rate.
>
> - We reran ACToR on the macro benchmark, with a strict safety requirement at each iteration, and removed the post-processing step. The results are consistent with the existing one: All Rust codes produced by ACToR are fully safe, and exhibit similar improvement in correctness compared with baseline. We moved the previous macro result to the Appendix.
>
> - We reported details of financial, token, and runtime costs for both micro- and macro-experiments. We added an equal-cost comparison to demonstrate the cost-effectiveness of ACToR.
>
> We use Claude Code with Claude-Sonnet-4.5 for the new and rerun experiments. Sonnet-4.5 has an identical cost as Sonnet-4 and completed the experiments noticeably faster in practice, enabling us to finish the additional experiments within the rebuttal timeframe. Importantly, our main claims and conclusions hold under Sonnet-4.5. The new results are consistent with those obtained using Sonnet-4.
>
> We have uploaded the revised paper and included a version highlighting the updates in the supplementary material.
>
> The answers to specific questions raised are below.

---

> ### Author Response · Authors · 2025-11-22
> **Author Response (Answer to Weaknesses 1/2)**
>
> #### **W1. The core comparisons are against a “naive” single-agent baseline and a “coverage” baseline. There is no head‑to‑head comparison with recent C to Rust systems that combine LLMs with analysis.**
>
> Our goal is to improve the correctness of fully automatic C to Rust translation that produces fully safe and runnable Rust programs, focusing on standalone command-line programs. As discussed in Sec 2.1, no prior C to Rust systems achieve this goal in a way that allows direct comparison without substantial human effort.
>
> Recent C to Rust LLM systems try to reduce unsafe portions in C2Rust translations [1,2] or translate from C code while incorporating test generation tools [3,4], decomposition strategies [5,10], dynamic pointer analysis [7], or static analysis [8,9]. However, their level of automation remains limited for generating correct whole-program translations. This is because these systems are built on vanilla LLMs with fixed workflow, which are brittle in out-of-design cases and thus lack full automation. So they may introduce human intervention into the system [7,8,10], adjust their goal to fragment-correctness or compilation [5,9], allow unsafe Rust [3,10], or struggle with long programs [4] (as evaluated in [6]).
>
> Our work targets whole-program correctness and generalizing to unseen tests under full automation via autonomous coding agents. As shown in the paper, single-agent approaches can’t give satisfactory results. Thus, we propose the GAN-style multi-agent design.
>
> A head-to-head comparison with systems that don’t target full safe Rust and whole program correctness is not apples-to-apples. For systems needing manual intervention, a fair and reproducible comparison across our 63 benchmarks is not feasible, as human effort varies widely.
>
> ---
>
> [1] Nitin, V., Krishna, R., Valle, L.L.D. and Ray, B., 2025. C2saferrust: Transforming c projects into safer rust with neurosymbolic techniques. arXiv preprint arXiv:2501.14257.
>
> [2] Sim, H., Cho, H., Go, Y., Fu, Z., Shokri, A. and Ravindran, B., 2025. Large Language Model-Powered Agent for C to Rust Code Translation. arXiv preprint arXiv:2505.15858.
>
> [3] Eniser, H.F., Zhang, H., David, C., Wang, M., Christakis, M., Paulsen, B., Dodds, J. and Kroening, D., 2024. Towards translating real-world code with llms: A study of translating to rust. arXiv preprint arXiv:2405.11514.
>
> [4] Yang, A.Z., Takashima, Y., Paulsen, B., Dodds, J. and Kroening, D., 2024. Vert: Verified equivalent rust transpilation with few-shot learning. arXiv preprint arXiv:2404.18852, 26.
>
> [5] Shiraishi, M. and Shinagawa, T., 2024. Context-aware code segmentation for c-to-rust translation using large language models. arXiv preprint arXiv:2409.10506.
>
> [6] Li, R., Wang, B., Li, T., Saxena, P. & Kundu, A., 2025, Translating C To Rust: Lessons from a User Study, Proceedings 2025 Network and Distributed System Security Symposium, Internet Society, San Diego, CA, USA.
>
> [7] Shetty, M., Jain, N., Godbole, A., Seshia, S.A. and Sen, K., 2024. Syzygy: Dual code-test c to (safe) rust translation using llms and dynamic analysis. arXiv preprint arXiv:2412.14234.
>
> [8] Zhou, T., Lin, H., Jha, S., Christodorescu, M., Levchenko, K. and Chandrasekaran, V., 2025. LLM-Driven Multi-step Translation from C to Rust using Static Analysis. arXiv preprint arXiv:2503.12511.
>
> [9] Ou, G., Liu, M., Chen, Y., Du, X., Wang, S., Zhang, Z., Peng, X. and Zheng, Z., 2025. Enhancing llm-based code translation in repository context via triple knowledge-augmented. arXiv preprint arXiv:2503.18305.
>
> [10] Bai, Y. and Palit, T., 2025. RustAssure: Differential Symbolic Testing for LLM-Transpiled C-to-Rust Code. arXiv preprint arXiv:2510.07604.

---

> ### Author Response · Authors · 2025-11-22
> **Author Response (Answer to Weaknesses 2/2)**
>
> #### **W2. The evaluation lacks runtime, token, and financial cost information, which is key for judging practicality at scale.**
>
> We added the cost information (line 371 and line 518) in the revised paper. The total cost of all evaluations in the paper is around 4000 USD. Under the ACToR setup, evaluating a single program in the macro benchmark costs approximately **27 USD**, using 30K input tokens, 57M cache-read tokens, and 317K output tokens on average. The end-to-end runtime is roughly 2.5 hours per program. Note that we ran multiple programs in parallel to reduce overall wall-clock time.
>
> ---
>
> #### **W3. Results appear to be single‑run without variance across seeds or model nondeterminism. Stability across runs is not reported. The text notes three programs that aborted before 10 iterations after multiple reruns, hinting at variability.**
>
> We have added the stability experiment (lines 470~474) for ACToR on the micro benchmark in the revised paper. We see only a minor standard deviation of **1.9** percentage points in pass rate across the 3 repeated trials in the 6 micro programs. Since repeated experiments incur significant cost, we are unable to afford running the stability experiment for every setup and configuration, and for all 57 macro programs.
>
> Comment for the unfinished iterations:
>
> We improved our implementation and made it more robust. **During the rerun of the macro experiment, all the programs completed 10 iterations**. The new results remain consistent with the original findings regarding safety and correctness improvements. This provides additional evidence for the stability of ACToR.
>
> ---
>
> #### **W4. It would be safer to qualify the claim with the particulars of the benchmark and evaluation protocol, or provide a stronger head‑to‑head against at least one recent approach on a shared subset.**
>
> We apologise if the writing was imprecise. We would like to clarify that in the revised paper: *"To our knowledge, it is the first work to show evidence that an agent-centric approach can reliably and automatically convert standalone command-line C programs at this scale”*. This matches our reported benchmarks precisely.

---

### Author Response · Authors · 2025-12-02
**Summary for Area Chairs and Reviewers**

As interactions with reviewers were disabled, we provide a brief summary for the area chairs and reviewers.

There were no significant objections in the initial reviews. The reviewers posed several important technical and conceptual questions. In response, we conducted **four** additional experiments and incorporated them into the revised paper. For the remaining conceptual questions, we addressed them in written responses. To our understanding, the additional evaluations, together with these responses, directly address the concerns raised in the reviews. The paper's main conclusions remain intact after the expanded evaluation.

Thank you once again for the time you devoted to reading our paper.

---

### Meta-Review · Area_Chair_FcpL · 2026-01-05

**Summary:**

Strengths pointed out by reviewers:
- The paper addresses a timely and challenging problem: translating nontrivial C codebases to safe Rust with minimal human intervention.
- ACToR consistently improves over the naive baseline across three agent-model choices.
- Meaningful contribution.
- Good ablation studies: the paper isolates the effect of adversarial test generation and a simple fuzzer

Weaknesses pointed out by reviewers:
- The core comparisons are against a “naive” single-agent baseline and a “coverage” baseline. There is no head‑to‑head comparison with recent C→Rust systems that combine LLMs with analysis. **Partially addressed.** It remains unclear if there could have been a simple combination (LLM with some form of static analysis) that would have provided a stronger baseline.
- The evaluation lacks runtime, token, and financial cost information, which is key for judging practicality at scale. **Addressed.**
- Results appear to be single‑run without variance across seeds or model nondeterminism. Stability across runs is not reported. The text notes three programs that aborted before 10 iterations after multiple reruns, hinting at variability. **Addressed.**
- The paper states “to our knowledge, it is the first such system that reliably translates C programs of this scale.” Given the breadth of recent efforts that blend LLMs with program analysis, it would be safer to qualify the claim with the particulars of the benchmark and evaluation protocol, or provide a stronger head‑to‑head against at least one recent approach on a shared subset. **Addressed.**
- Unsound problem definition. The authors' definition of C-to-Rust translation does not require exclusive use of safe features, yet claims that "memory safe handling of malicious inputs is guaranteed by the Rust compiler itself." Since unsafe code may be involved, this assumption does not hold - it is entirely possible for inputs outside the valid universe U to expose vulnerabilities on the Rust side, which may be the same as or different from those on the C side, and which, at worst, are new vulnerabilities altogether. As such scenarios are not considered in the behavior equivalence check, such a problem definition (and any of its solutions) would defeat the purpose of what motivates translating from C to Rust in the first place (though ACToR apparently does not literally address this problem definition, as it is said to enforce safe Rust as a post-processing step). **Addressed.**
- Unprincipled experimental configuration. The decision to use 10 turns and 3 tests each turn feels handwavy. It seems entirely likely that with more turns and more tests added, more divergences between the C code and Rust code would be exposed. While I understand that the authors are limited by the cost of running the LLMs, I think it is very worthwhile to experimentally study the trade-offs between various (# turns, # tests) configurations and how they impact the degree of translation correctness ACToR can achieve. **Partially addressed.** More ablations and some illustration to visualize the trends would have been nice as it might be worth for future work to spend much more to improve results.
- Correctness is proxied by tests. However, tests may not be able to expose real bugs. No human validation of the translated code or the quality / sufficiency of the test cases. **Partially addressed.** No human validation.

Questions asked by reviewers:
- How well would this method generalize to other existing datasets, e.g., CRUST-Bench: A Comprehensive Benchmark for C-to-safeRust Transpilation? **Partially addressed.** The authors did not discuss if tests for the library could be used to make ACToR work.
- Is there any guarantee that the final generated Rust programs are really memory-safe? Or the agents could also escape certain checks to bypass? How much confidence in correctness can you get out of the generated Rust programs? Is there any manual validation? **Addressed.**
- What if you only use the fuzzing script without the agent? **Partially addressed** It would have been nice to have a small evaluation.
- Can you compare your approach with the other paper, "Exploring and Unleashing the Power of Large Language Models in Automated Code Translation", and tell me what your technical novelty is? Similarly, for this paper, "AlphaTrans: A Neuro-Symbolic Compositional Approach for Repository-Level Code Translation and Validation". **Addressed.**

**Reviewer Concerns:**

See above.

**Reviewer Scores:**

- Reviewer vkya: $4 \to 6$
- Reviewer hYZH: $6 \to 6$
- Reviewer NQvr: $4 \to 4$

---

### Decision · Program_Chairs · 2026-01-26

Reject